# Development of Artificial Intelligence-Based Dual-Energy Subtraction for Chest Radiography

**Asumi Yamazaki [1,†], Akane Koshida [1,†], Toshimitsu Tanaka [2], Masashi Seki [3] and Takayuki Ishida [1,*]**

1   Division of Health Sciences, Graduate School of Medicine, Osaka University, Suita 565-0871, Japan; yamazaki-a@sahs.med.osaka-u.ac.jp (A.Y.)
2   Department of Radiology, National Cerebral and Cardiovascular Center, Suita 564-8565, Japan
3   Department of Radiology, Kitasato University Hospital, Sagamihara 252-0329, Japan
*   Correspondence: tishida@sahs.med.osaka-u.ac.jp; Tel.: +81-6-6879-2573
†   These authors contributed equally to this work.

**Abstract:** Recently, some facilities have utilized the dual-energy subtraction (DES) technique for chest radiography to increase pulmonary lesion detectability. However, the availability of the technique is limited to certain facilities, in addition to other limitations, such as increased noise in high-energy images and motion artifacts with the one-shot and two-shot methods, respectively. The aim of this study was to develop artificial intelligence-based DES (AI–DES) technology for chest radiography to overcome these limitations. Using a trained pix2pix model on clinically acquired chest radiograph pairs, we successfully converted 130 kV images into virtual 60 kV images that closely resemble the real images. The averaged peak signal-to-noise ratio (PSNR) and structural similarity (SSIM) between virtual and real 60 kV images were 33.8 dB and 0.984, respectively. We also achieved the production of soft-tissue- and bone-enhanced images using a weighted image subtraction process with the virtual 60 kV images. The soft-tissue-enhanced images exhibited sufficient bone suppression, particularly within lung fields. Although the bone-enhanced images contained artifacts on and around the lower thoracic and lumbar spines, superior sharpness and noise characteristics were presented. The main contribution of our development is its ability to provide selectively enhanced images for specific tissues using only high-energy images obtained via routine chest radiography. This suggests the potential to improve the detectability of pulmonary lesions while addressing challenges associated with the existing DES technique. However, further improvements are necessary to improve the image quality.

**Keywords:** dual-energy subtraction; chest radiography; artificial intelligence; deep learning; pix2pix

## 1. Introduction

Lung cancer is the disease with the highest mortality and the second-highest incidence of cancer worldwide [1,2]. Since early-stage lung cancer may have a better prognosis with appropriate treatment, early diagnosis and accurate staging are critical [2,3]. Randomized controlled trials, including the National Lung Screening Trial (NLST), have demonstrated that the use of low-dose computed tomography (CT) for lung cancer screening reduces the mortality by 20% compared to chest radiography [4,5]. Chest radiography lacks effectiveness [4], and the American Cancer Society Lung Cancer Screening Guidelines recommend low-dose CT rather than chest radiography [6]. Nevertheless, due to the low cost, low radiation dose, and high adoption rate of the equipment, chest radiography is widely performed for lung cancer screening, in addition to low-dose CT [7–9].

Additionally, some facilities utilize the dual-energy subtraction (DES) technique for chest radiography [10]. This technique can produce images that emphasize tissues with particular linear attenuation coefficients [11] and typically produces soft-tissue-enhanced and bone-enhanced images [12]. It has been reported that soft-tissue-enhanced images can

improve the ability to detect pulmonary lesions [13–16]. Oda et al., compared the performance of radiologists in detecting pulmonary lesions by clinical chest radiography with and without the DES technique, and their receiver operating characteristic (ROC) analysis demonstrated the statistically significant superiority of using DES images [15]. Manji et al., reported that soft-tissue-enhanced images obtained through the DES technique statistically significantly reduced the reading time of radiologists and slightly improved the diagnostic accuracy of pulmonary lesions [16]. The superiority of soft-tissue-enhanced images obtained via the DES technique in the diagnosis of COVID-19 has been also confirmed [17]. Furthermore, research on advanced DES techniques, such as the optimization of exposure conditions [18] and the automatic determination of weight factors for bone and soft tissue enhancements in the subtraction process [19], has been actively reported.

However, such a DES technique has some problems. First, only limited numbers of facilities own the necessary systems. Second, there are several problems associated with the imaging techniques of the one-shot and two-shot methods. In the one-shot method, two images are simultaneously obtained at different energies by placing a thin copper plate between two imaging detectors [20]. Because of the copper plate, noise characteristics can deteriorate the quality of high-energy images [21]. In the two-shot method, on the other hand, X-ray exposure is carried out twice at different energies [22]. As a result, motion artifacts and dose increments are unavoidable.

Therefore, we aim to address these issues by developing a technique to virtually generate low-energy images from high-energy images using artificial intelligence (AI). This development of AI-based DES (AI–DES) does not require a specific imaging detector with a metal plate or multiple exposures. Moreover, the AI-DES can arbitrarily select the enhanced tissue by adjusting the weight of the image subtraction process. Consequently, our AI-DES has the potential to provide more enriched information than existing methods, where bone-suppressed images are directly produced [23–31]. For instance, Liu et al., developed an AI model to generate DES-like soft tissue images, but their approach primarily focused on suppressing bone tissues, making it difficult to selectively enhance specific tissues [25]. Similarly, Bae et al., developed a generative adversarial network (GAN)-based bone suppression model for chest radiography, and demonstrated that its ability to detect pulmonary lesions is comparable to that of a DES technique [26]. Cho et al., achieved bone suppression on pediatric chest radiographs by utilizing computed tomographic images of adults and pediatrics to train the AI model [27]. In contrast, our AI-DES, by adapting weighted image subtraction with artificially synthesized low-energy images, has a significant advantage in terms of generating not only soft-tissue- or bone-enhanced images but also selectively enhanced images of tissues with a specific linear attenuation coefficient. However, this paper mainly focuses on generating soft-tissue- and bone-enhanced images for comparison with existing DES systems as an initial report on the development of AI-DES.

We employed pix2pix [32], which is a well-established image-to-image translation network, to construct the AI network. Pix2pix has been widely used in many image domain transformation tasks, such as image colorization and style transfer. It is also extensively used for medical imaging. Yoshida et al., adopted pix2pix to correct motion artifacts in magnetic resonance (MR) images [33]. Sun et al., achieved denoising of low-dose single-photon emission-computed tomography (SPECT) images using pix2pix [34]. Although pix2pix usually requires identical positional information between the paired images, our task of generating low-energy images from high-energy input images can deal with this constraint.

The main contributions of this work are as follows:

1. We developed an AI-based DES system to provide soft-tissue- and bone-enhanced images using virtually generated low-energy images;
2. The virtual low-energy images were generated through the AI technique from only high-energy images, which can be obtained by routine chest radiography;
3. AI-DES has the potential to provide specific tissue-enhanced images while avoiding issues associated with DES systems, such as multiple exposures and noise increments;

4.   A comparison of the generated images with those produced by a clinically applied system suggests that AI-DES can achieve superior sharpness and noise characteristics.

Furthermore, although this is a future-expanded perspective, the novelty of the AI-DES is that it allows for the selection of enhanced tissues by adjusting the weight in the image subtraction process, in comparison to existing works.

In Section 2, we first introduce the developed AI-DES system. Then, we describe image datasets and prepossessing steps for AI training. Next, we specify the training setups and explain the methods used to evaluate similarity between the generated and ground truth images. Section 3 presents the generated images in comparison to the ground truth. Section 4 discusses the performance, limitations, and future perspectives of our AI-DES. Finally, Section 5 summarizes and concludes this work.

## 2. Materials and Methods

### 2.1. AI-DES Development

Our developed AI-DES consists of an AI network and a weighted image subtraction process. We first describe the AI network, then explain the image subtraction process.

#### 2.1.1. AI Network

We employed pix2pix for our AI network to convert high-energy images into low-energy images. This is a variant of a conditional generative adversarial network (cGAN) [35]. Unlike a typical cGAN that produces images from a random noise vector, the generator of pix2pix takes images as input and transforms them into images of a different domain by learning the relationship between the two domains [32]. The discriminator receives an image pair of two domains and attempts to determine whether the pair is real or fake.

Figure 1 illustrates the pix2pix network used in this study. The generator learns to convert high-energy images into virtual low-energy images similar to the corresponding real low-energy images. Simultaneously, the discriminator aims to distinguish between the pair of real high-energy and virtual low-energy images and the pair of real high-energy and low-energy images. The learning process is expressed as a min–max game with an adversarial loss function given by

$$\min_G \max_D L_{cGAN}(G,D) = \mathop{\mathbb{E}}_{x \sim \mathbb{P}_{high}, y \sim \mathbb{P}_{low}} [\log D(x,y)] + \mathop{\mathbb{E}}_{x \sim \mathbb{P}_{high}} [\log(1 - D(x, G(x)))], \quad (1)$$

where $x \sim \mathbb{P}_{high}$ represents high-energy images, $y \sim \mathbb{P}_{low}$ represents low-energy images, $G$ is the generator, and $D$ is the discriminator [34]. $G$ attempts to minimize Equation (1), while $D$ attempts to maximize it.

Pix2pix also imposes a constraint on the L1 distance between the generated and real images to make the generator produce images that are closer to the ground truth, as follows:

$$L_1(G) = \mathop{\mathbb{E}}_{x \sim \mathbb{P}_{high}, y \sim \mathbb{P}_{low}} [||y - G(x)||_1]. \quad (2)$$

Thus, the objective function of pix2pix can be expressed as:

$$G^* = \min_G \max_D L_{cGAN}(G,D) + \lambda L_1(G), \quad (3)$$

where $\lambda = 100$ was used in this study.

We implemented the pix2pix network by modifying a publicly available code [36]. Specifically, we changed the resolution of each layer of the generator and discriminator to produce images with a resolution of $1024 \times 1024$. Otherwise, the same architecture as the original pix2pix [32] was used. The details are described below and in Section 2.3.

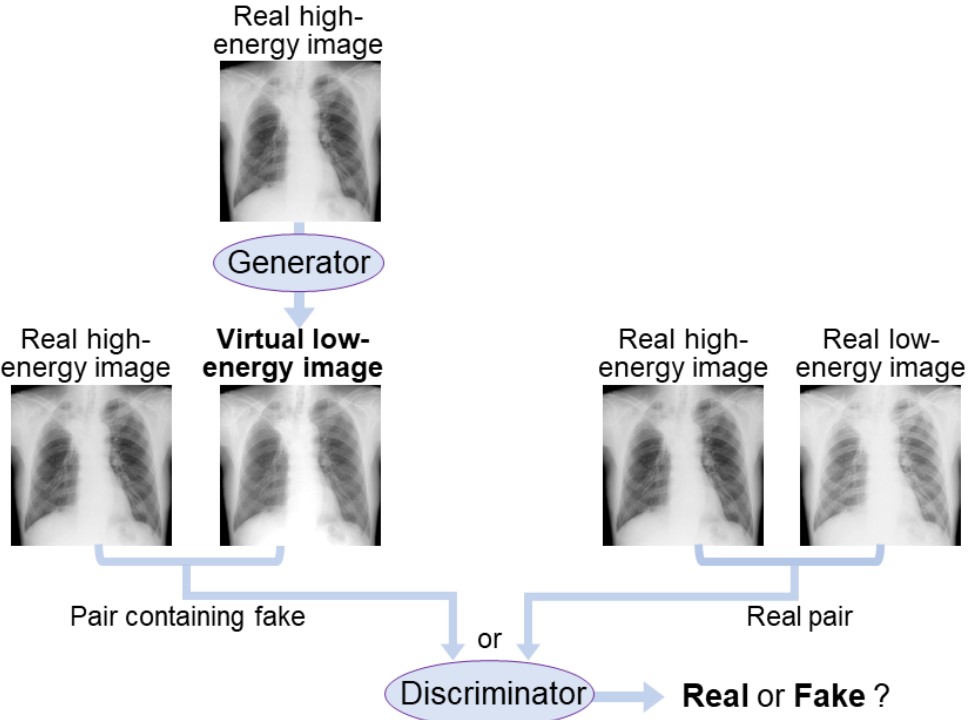

**Figure 1.** AI network diagram using pix2pix. The generator attempts to produce virtual low-energy images that resemble the real low-energy images from high-energy images. The discriminator aims to distinguish between the pairs containing virtual images and the real image pairs.

Figure 2 and Table 1 show the generator network architecture in AI-DES. The generator has a 16-layer U-Net [37] structure with symmetric encoder and decoder parts. The encoder part has eight layers, consisting of two-dimensional convolution (Conv2d: kernel size = 4, stride = 2, padding = 1), batch normalization (BN), and a LekyReLU activation function. The decoder part also has eight layers, consisting of two-dimensional deconvolution (Deconv2d: kernel size = 4, stride = 2, padding = 1), BN, and a ReLU activation function. Dropout layers were inserted between the first and second, second and third, and third and fourth layers of the decoder part. The tanh activation function was applied to the final layer to output the virtual low-energy images. Most importantly, the U-Net architecture has skip connections that concatenate the mirrored encoder and decoder layers to recover high-frequency components.

The discriminator has five convolutional neural network (CNN) layers, as shown in Figure 3 and Table 2. The first layer takes six channels, since the discriminator receives a pair of two images, each with three channels. The first through third layers downsample the feature maps using Conv2d (kernel size = 4, stride = 2, padding = 1), while the fourth and final layers reduce the map resolution by one pixel using Conv2d (kernel size = 4, stride = 1, padding = 1). The discriminator employs the PatchGAN [38] approach to evaluate multiple image patches and averages the loss over the output map ($126 \times 126 \times 1$) to distinguish between real and virtual images.

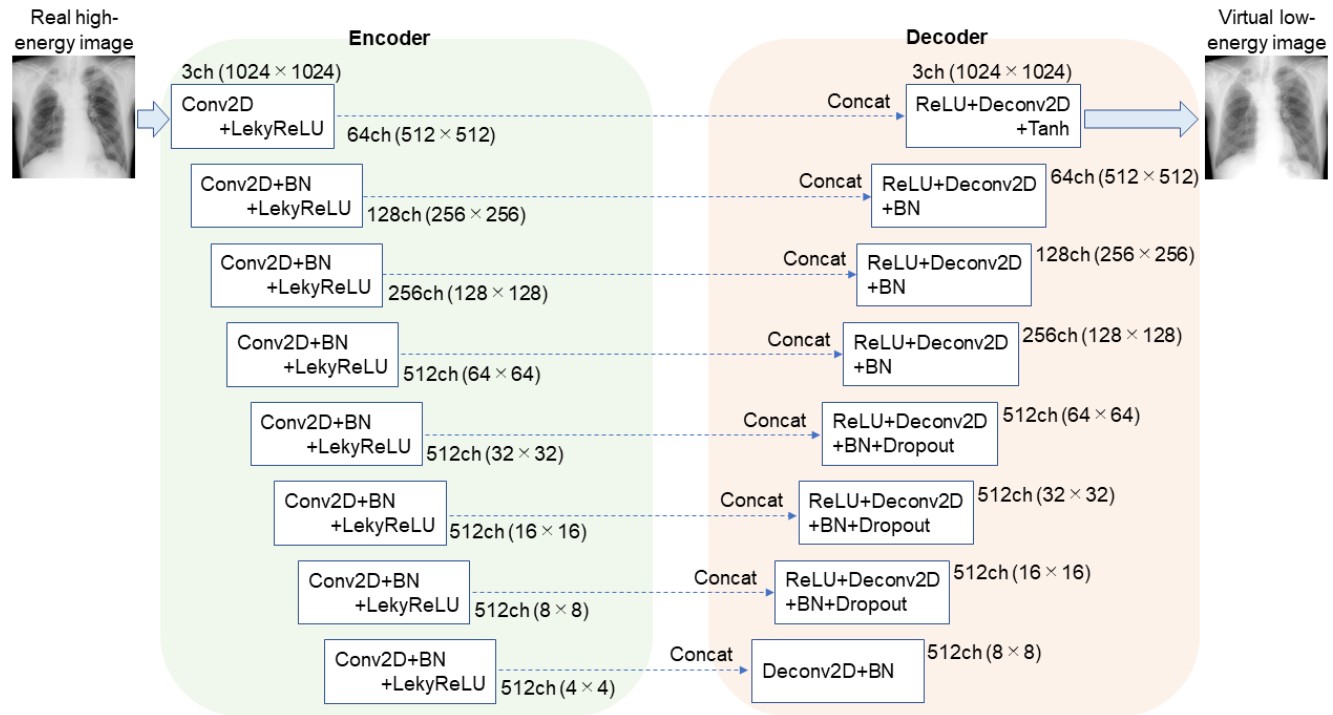

**Figure 2.** Generator network used in AI-DES. The network has a 16-layer U-Net structure. The skip connections concatenate the mirrored encoder and decoder layers to recover high-frequency components in the generated image quality.

**Table 1.** Generator architecture in AI-DES.

|  |  | Type | Norm [1], Dropout | Activation | Input Shape [2] | Output Shape [2] |
|---|---|---|---|---|---|---|
| Encoder | Layer1 |  | – |  | $1024 \times 1024 \times 3$ | $512 \times 512 \times 64$ |
|  | Layer2 | Conv2d (4,2,1) | BN | LekyReLU | $512 \times 512 \times 64$ | $256 \times 256 \times 128$ |
|  | Layer3 |  |  |  | $256 \times 256 \times 128$ | $128 \times 128 \times 256$ |
|  | Layer4 |  |  |  | $128 \times 128 \times 256$ | $64 \times 64 \times 512$ |
|  | Layer5 |  |  |  | $64 \times 64 \times 512$ | $32 \times 32 \times 512$ |
|  | Layer6 |  |  |  | $32 \times 32 \times 512$ | $16 \times 16 \times 512$ |
|  | Layer7 |  |  |  | $16 \times 16 \times 512$ | $8 \times 8 \times 512$ |
|  | Layer8 |  |  |  | $8 \times 8 \times 512$ | $4 \times 4 \times 512$ |
| Decoder | Layer9 | Deconv2d (4,2,1) | BN |  | $4 \times 4 \times 512$ | $8 \times 8 \times 512$ |
|  | Layer10 | ReLU+Deconv2d (4,2,1) | BN+Dropout | – | $8 \times 8 \times 512$ | $16 \times 16 \times 512$ |
|  | Layer11 |  |  |  | $16 \times 16 \times 512$ | $32 \times 32 \times 512$ |
|  | Layer12 |  |  |  | $3 2 \times 32 \times 512$ | $64 \times 64 \times 512$ |
|  | Layer13 |  | BN |  | $64 \times 64 \times 512$ | $128 \times 128 \times 256$ |
|  | Layer14 |  |  |  | $128 \times 128 \times 256$ | $256 \times 256 \times 128$ |
|  | Layer15 |  |  |  | $256 \times 256 \times 128$ | $512 \times 512 \times 64$ |
|  | Layer16 |  | – | Tanh | $512 \times 512 \times 64$ | $1024 \times 1024 \times 3$ |

[1] Normalization. [2] Width × height × channel.

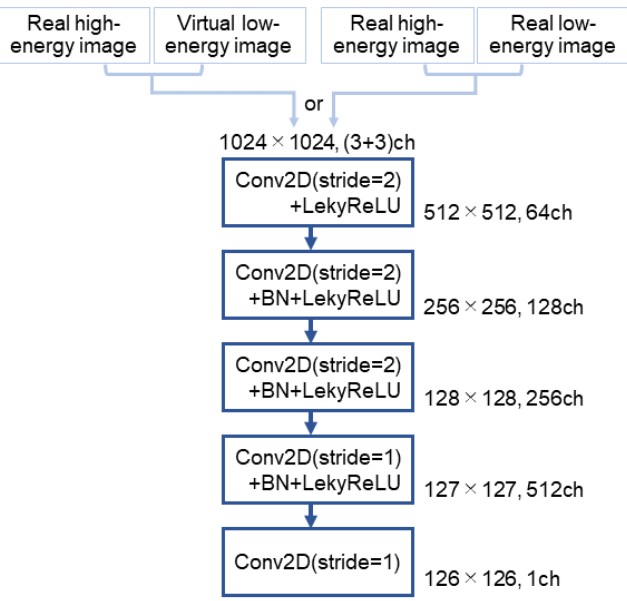

**Figure 3.** Discriminator network used in AI-DES. The network comprises five convolutional neural network layers. It takes either a pair of real high-energy and virtual low-energy images or a pair of real high-energy and low-energy images as input . The network then outputs a feature map that determines whether the pair is real or fake.

**Table 2.** Discriminator architecture in AI-DES.

|        | Type            | Normalization | Activation | Input Shape [1]            | Output Shape [1]           |
|--------|-----------------|---------------|------------|----------------------------|----------------------------|
| Layer1 |                 | –             |            | $1024 \times 1024 \times 6$ | $512 \times 512 \times 64$  |
| Layer2 | Conv2d (4,2,1)  |               | LekyReLU   | $512 \times 512 \times 64$  | $256 \times 256 \times 128$ |
| Layer3 |                 | BN            |            | $256 \times 256 \times 128$ | $128 \times 128 \times 256$ |
| Layer4 | Conv2d (4,1,1)  |               |            | $128 \times 128 \times 256$ | $127 \times 127 \times 512$ |
| Layer5 |                 | –             | –          | $127 \times 127 \times 512$ | $126 \times 126 \times 1$   |

[1] Width × height × channel.

### 2.1.2. Weighted Image Subtraction

We assumed that raw data of monochromatic low- and high-energy images in direct-conversion flat-panel detector (d-FPD) systems have pixel values of $P_L$ and $P_H$, respectively as expressed by [11,39]

$$\log_{10}(P_L) = -(\mu_B(L) \cdot t_B + \mu_S(L) \cdot t_S), \tag{4}$$

$$\log_{10}(P_H) = -(\mu_B(H) \cdot t_B + \mu_S(H) \cdot t_S), \tag{5}$$

where $\mu_B$ and $\mu_S$ are the linear attenuation coefficients of bone and soft tissues at low ($L$) or high ($H$) energy, respectively; $t_B$ and $t_S$ denote the thicknesses of bone and soft tissues, respectively; and $P_L$ and $P_H$ are proportional to low- and high-energy X-ray intensity transmitted through the tissues, respectively, since d-FPD systems have a linear response to X-ray intensity. The weighted subtraction of Equations (4) and (5) is given by:

$$K_H \cdot \log_{10}(P_H) - K_L \cdot \log_{10}(P_L) = (K_L \cdot \mu_B(L) - K_H \cdot \mu_B(H))t_B + (K_L \cdot \mu_S(L) - K_H \cdot \mu_S(H))t_S, \tag{6}$$

where $K_L$ and $K_H$ are weight factors.

When $(K_L \cdot \mu_B(L) - K_H \cdot \mu_B(H))$ equals zero, Equation (6) represents the emphasized difference in logarithmically amplified data of X-ray intensity transmitted through soft

tissues at low and high energies. Finally, it corresponds to the pixel values ($P_{sub}$) of the specific tissue-enhanced images, as expressed by

$$P_{sub} = K_H \cdot \log_{10}(P_H) - K_L \cdot \log_{10}(P_L). \tag{7}$$

Apart from the raw data, such as $P_L$ and $P_H$, $P_{sub}$ denotes the pixel values for viewing, where log-amplified X-ray intensity distribution is exhibited.

Here, the weight factor ($\omega_S$) for soft-tissue-enhanced image generation is given by:

$$\omega_S = \frac{K_H}{K_L} = \frac{\mu_B(L)}{\mu_B(H)}. \tag{8}$$

Similarly, the weight factor ($\omega_B$) for bone-enhanced images is given by:

$$\omega_B = \frac{K_H}{K_L} = \frac{\mu_S(L)}{\mu_S(H)}. \tag{9}$$

It should be noted that $\omega_S$ and $\omega_B$ in Equations (8) and (9) represent theoretical values. In this study, we obtained the $P_{sub}$ of soft-tissue- and bone-enhanced images by using the raw data of real high-energy and virtual low-energy images, as expressed by

$$P_{sub} = \omega \cdot \log_{10}(P_H) - \log_{10}(P_L), \tag{10}$$

where $\omega$ is a weight factor. We adjusted the value of $\omega$ for each test case to most selectively emphasize the target tissues. We focused on generating soft-tissue- and bone-enhanced images to compare the performance with that of an existing DES system. However, the AI-DES can provide the option of arbitrarily targeting specific tissues for enhancement by adjusting the weight factor.

### 2.2. Dataset Preparation

We used raw data of chest radiographs taken by a clinically applied two-shot DES system (Discovery XR656, GE Healthcare, Chicago, IL, USA) at Kitasato University Hospital (Sagamihara City, Japan) to create our datasets. The tube voltages were 130 kV for high-energy images and 60 kV for low-energy images. The imaging detector, consisting of amorphous silicon with a cesium iodide (CSI) scintillator, is a d-FPD type with $3524 \times 4288$-pixel arrays. The total number of cases was 300.

We first cropped Digital Imaging and Communications in Medicine (DICOM)-formatted images with a 12 bit contrast resolution to $2022 \times 2022$ pixels centered on the lung region. The images were converted into tagged image file format (TIFF) images with $1024 \times 1024$ pixels using the bilinear interpolation method. The image pair consisting of 130 kV and 60 kV images from each patient was input to AI–DES after being normalized to a range of 0–1 for training. We used 240 pairs of images for training, 30 pairs for validation, and 30 pairs for testing. ImageJ software (1.53e, National Institutes of Health, Bethesda, MD, USA) was used to set up these datasets.

### 2.3. Training Environment and Parameter Settings

We used an Intel Core (TM) i7-9700K CPU and an NVIDIA GeForce RTX 2080 with 8 GB GPU memory for training. GPU acceleration was enabled using CUDA version 10.0.130, and cuDNN version 7.4.1.5-1+cuda10.0 was utilized. The implementation was performed using Python 3.7.10 and the PyTorch 1.10.0 framework on an Ubuntu 18.04.4 LTS operating system. We set the maximum number of epochs to 4000 and the batch size to 2. Adam optimization was used with the following momentum parameters: $\beta_1 = 0.5$ and $\beta_2 = 0.999$. We dynamically adjusted the learning rates as the training progressed. The learning rate of the generator started at 0.002 and decreased linearly by 0.002/4000 per epoch. The learning rate of the discriminator also decreased linearly by 0.02/4000 per epoch, starting from 0.02.

*2.4. Performance Evaluation*

We evaluated similarity between real and virtually generated 60 kV images for the test dataset cases. In addition, image quality of soft-tissue- and bone-enhanced images generated by our AI-DES was evaluated based on their similarity to those obtained by Discovery XR656, which is assumed to be the ground truth. Frèchet inception distance (FID) has been widely used for performance evaluation of GAN models [40,41]. However, this metric measures distances between synthetic and real data distributions; that is, FID evaluates not the similarity between each real and fake sample but the entire similarity between the two groups. Hence, we evaluated the similarity between each generated image and its corresponding ground truth using the peak signal-to-noise ratio (PSNR), structural similarity (SSIM) [42,43], and multiscale SSIM (MS-SSIM) [44] instead of FID.

PSNR is an index based on the perceived sensitivity of noise components. It calculates the noise ratio relative to the maximum value between two images in decibels, as follows:

$$PSNR = 20 \log_{10} \left( \frac{P_{max}}{MSE} \right),$$ (11)

where MSE is the mean square error (MSE) between two images, and $P_{max}$ is the maximum value of the image pixels. In this study, $P_{max}$ was set to 1 because we normalized the pixel values of each image dataset, as mentioned previously in Section 2.3. The higher the value of PSNR, the more similar the two images are.

SSIM assumes image similarity using three components of brightness, contrast, and structure [42]. It has been reported that SSIM is more consistent with human perception and subjective evaluation than PSNR [45]. The images become more similar when the SSIM value is closer to 1. SSIM is calculated by dividing each region of interest (ROI) and averaging the respective SSIM values to estimate the overall similarity between the two entire images (**x**,**y**). We set the ROI size to 3 × 3 in this study. The calculation of SSIM between two corresponding ROIs is defined as follows:

$$SSIM_{ROI}(\mathbf{x}, \mathbf{y}) = \frac{(2\mu_x\mu_y + C_1)(2\sigma_{xy} + C_2)}{(\mu_x^2 + \mu_y^2 + C_1)(\sigma_x^2 + \sigma_y^2 + C_2)},$$ (12)

where $\mu_x$ and $\mu_y$ are the local averages, $\sigma_x$ and $\sigma_y$ are the local standard deviations, and $\sigma_{xy}$ is local covariance. $C_1$ and $C_2$ are given by

$$C_1 = (K_1 L)^2,$$ (13)

$$C_2 = (K_2 L)^2,$$ (14)

where $K_1$ = 0.01, $K_2$ = 0.03, and $L$ = 1 were used in this study.

MS-SSIM has been introduced as an alternative metric of SSIM to evaluate image details at various resolutions [44]. It can overcome the shortcomings of SSIM, which tends to underestimate spatial translation and overestimate image blurring [46]. MS-SSIM is computed by combining the three components of SSIM on multiple scales, as follows:

$$SSIM(\mathbf{x}, \mathbf{y}) = [l_M(\mathbf{x}, \mathbf{y})]^{\alpha_M} \cdot \prod_{j=1}^{M} [c_j(\mathbf{x}, \mathbf{y})]^{\beta_j} [s_j(\mathbf{x}, \mathbf{y})]^{\gamma_j}.$$ (15)

The two images (**x**,**y**) are iteratively low-pass-filtered and downsampled by a factor of 2. The scale of the original image is 1, while that of the most reduced image is $M$. The brightness is compared only at scale $M$, and we refer to this as $l_M(\mathbf{x}, \mathbf{y})$. The contrast and structure components are compared at each scale, denoted as $c_j(\mathbf{x}, \mathbf{y})$ and $s_j(\mathbf{x}, \mathbf{y})$, respectively, for the $j$th scale. Wang et al. [44] obtained five-scale parameters in which the SSIM scores agreed with subjective assessments: $\beta_1 = \gamma_1 = 0.0448$, $\beta_2 = \gamma_2 = 0.2856$,

$\beta_3 = \gamma_3 = 0.3001$, $\beta_4 = \gamma_4 = 0.2363$, and $\alpha_5 = \beta_5 = \gamma_5 = 0.1333$. We set $M = 5$ and used these five-scale parameters in the present study.

We also subjectively evaluated the image quality of the soft-tissue- and bone-enhanced images generated by AI-DES in comparison to those obtained using Discovery XR656. The subjective evaluation was conducted by three authors (A.Y., A.K., and T.I.), who are all radiological technologists, to determine whether the both images of identical patients looked similar.

## 3. Results

### 3.1. Generated Virtual Low-Energy Images

The real and 60 kV images virtually generated by our AI network for four test cases are presented in Figure 4a–d. The real and generated images look quite similar to each other. The calculated PSNR, SSIM, and MS-SSIM values for each case are shown at the bottom of the figure. The average PSNR, SSIM, and MS-SSIM values across all test cases were 33.8 dB, 0.984, and 0.957, respectively.

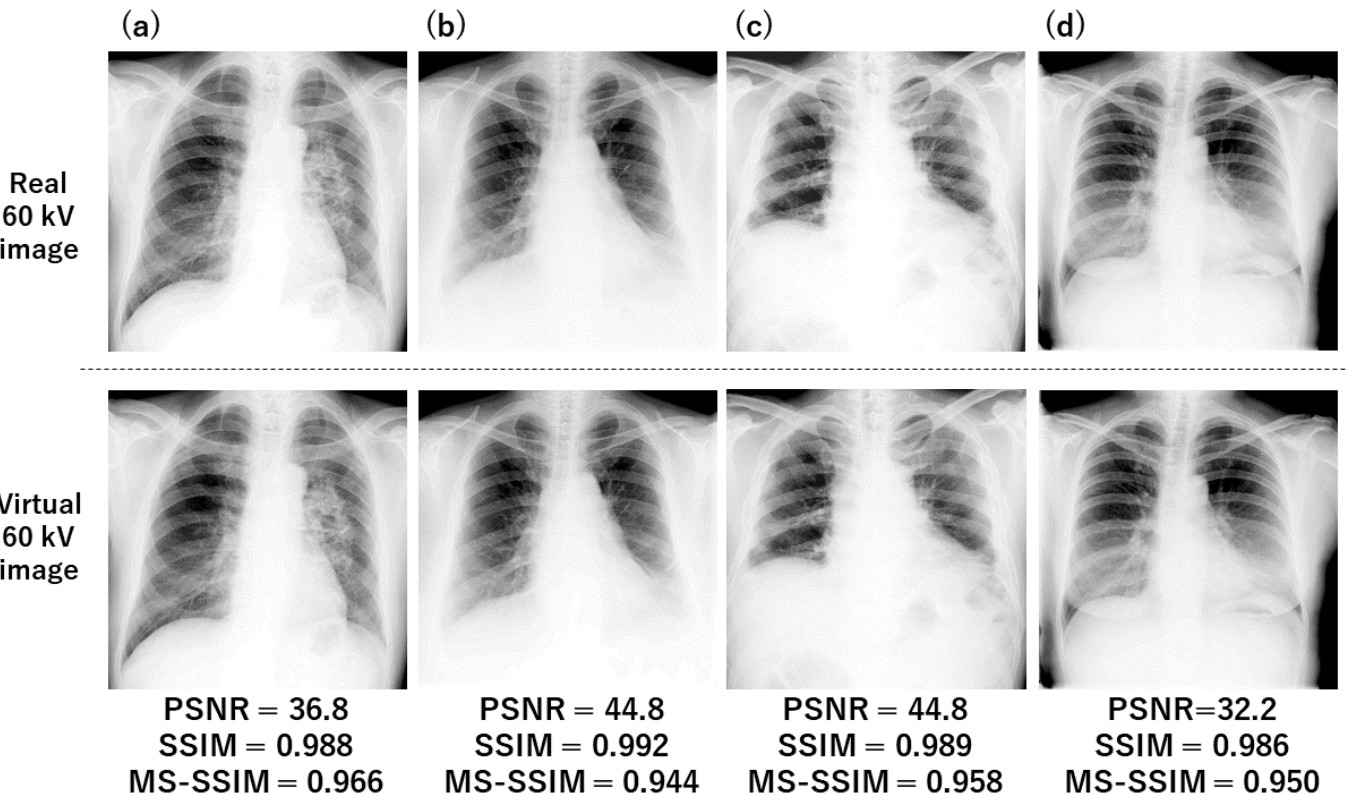

**Figure 4.** Examples of low-energy images virtually generated by our trained AI network. Four test cases (**a**–**d**) are presented here. The similarity indices between real and virtual images are presented at the bottom of each figure.

### 3.2. Soft Tissue and Bone Images

Figure 5 shows soft-tissue- and bone-enhanced images generated using Equation (10) for four test cases. The weight factors ($\omega$) used in the weighted subtraction are presented in the lower-right corner of each image. As shown in the left column in Figure 5, the soft-tissue-enhanced images demonstrate that the soft tissues were well-retained, and bone tissues were effectively suppressed, although the edges of the clavicles, ribs, and spine are faintly presented. On the other hand, as seen in the right column in Figure 5, the bone-enhanced images exhibited relatively suppressed soft tissues. However, the enhanced lower thoracic and lumbar spine are poorly visualized, as they appear to be blacked-out.

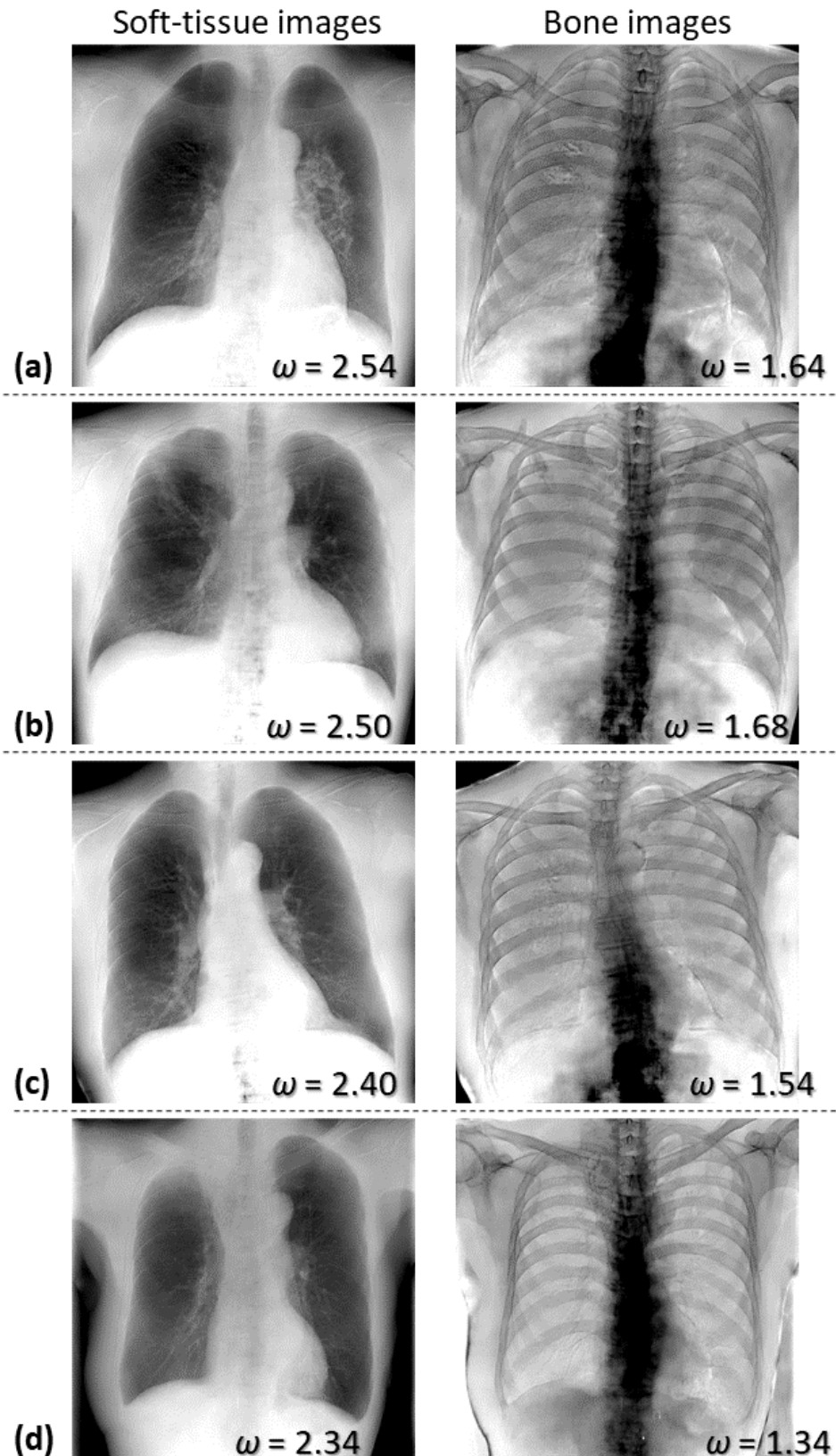

**Figure 5.** Examples of soft-tissue- and bone-enhanced images generated by AI-DES for four test cases (**a**–**d**). The weight factors used in the subtraction process are presented in the lower-right corner of each image.

Table 3 presents the average and standard deviation of PSNR, SSIM, and MS-SSIM values across all test cases, evaluating the similarity between soft-tissue-enhanced images obtained using AI-DES and Discovery XR656, as well as the similarity between bone-enhanced images. Additionally, Table 3 includes weight factors used in weighted image subtraction of AI-DES, as well as the similarity indices between the real and virtual 60 kV images, as explained in Section 3.1. The PSNR, SSIM, and MS-SSIM values for soft-tissue- and bone-enhanced images are significantly lower than the indices for 60 kV images. Table 3 also demonstrates that the values of $\omega$ varied slightly among patients, but $\omega$ values for soft tissue enhancements were consistently higher than those for bone enhancements.

**Table 3.** Similarity indices (average $\pm$ standard deviation) for all test cases and weight factors used to produce soft-tissue- and bone-enhanced images in AI-DES.

|  | PSNR | SSIM | MS-SSIM | Weight Factor |
|---|---|---|---|---|
| 60 kV images (virtual and real) | $33.8 \pm 5.39$ | $0.984 \pm 0.00554$ | $0.957 \pm 0.0514$ | – |
| Soft tissue images (AI-DES and Discovery) | $21.1 \pm 2.56$ | $0.711 \pm 0.0551$ | $0.794 \pm 0.0640$ | $2.47 \pm 0.159$ |
| Bone images (AI-DES and Discovery) | $18.3 \pm 1.97$ | $0.433 \pm 0.0827$ | $0.571 \pm 0.101$ | $1.52 \pm 0.102$ |

Figures 6 and 7 compare our soft-tissue- and bone-enhanced images, which were generated from the real 130 kV and virtual 60 kV images, to those obtained using Discovery XR656 for the respective cases. The weight factors and calculated PSNR, SSIM, and MS-SSIM values are also shown for the soft-tissue- and bone-enhanced images. While the Discovery system uses real 60 kV images in the subtraction process, our AI-DES utilizes virtually generated 60 kV images. Not only in these two cases but most test cases, the soft-tissue-enhanced images demonstrated that the bone shadows within the lung fields were successfully suppressed in both systems. However, our soft-tissue-enhanced images contained artifacts, implying the presence of thoracic and lumbar spines. Furthermore, in some cases, the mediastinum or liver area appeared overly bright in our soft-tissue-enhanced images when adjusting the contrast within the lung fields to match that in the images produced by the Discovery system, as particularly seen in the case of Figure 7.

Subsequently, as compared in Figures 6 and 7, we confirmed that the bone images produced by the Discovery system exhibited remarkably more selective enhancement for bone tissues across the entire image. In contrast, our bone-enhanced images presented comparably enhanced ribs but contained artifacts in and around the region where the lower thoracic and lumbar spines should be depicted.

Figures 8 and 9 also compare the soft-tissue- and bone-enhanced images generated by AI-DES to those generated by the Discovery system for other test cases, as well as enlarged views of specific areas. The enlarged views revealed that the images generated by AI-DES (the upper row in Figures 8b and 9b) exhibited superior sharpness and noise characteristics, particularly in the bone-enhanced images, compared to those generated by the Discovery system (the lower row in Figures 8b and 9b). Alternatively, the soft-tissue-enhanced images generated by the Discovery system showed better contrast, particularly in depicting pulmonary vessels and soft tissue lesions, and more effectively suppressed bone shadows.

Taken together, our bone suppression within the lung fields was relatively successful, although the similarity indices were not substantially high. In other words, AI-DES was able to selectively enhance soft tissues, especially within lung fields, comparable to the existing DES system using only high-energy images. Nonetheless, bone edge artifacts and excessive contrast were exhibited in the mediastinum and liver areas. The red arrows in Figures 6 and 7 indicate pulmonary lesions. Although these lesions can already be easily observed in the real 130 kV images, both AI-DES and the Discovery system successfully enhanced the lesions in the soft-tissue-enhanced images.

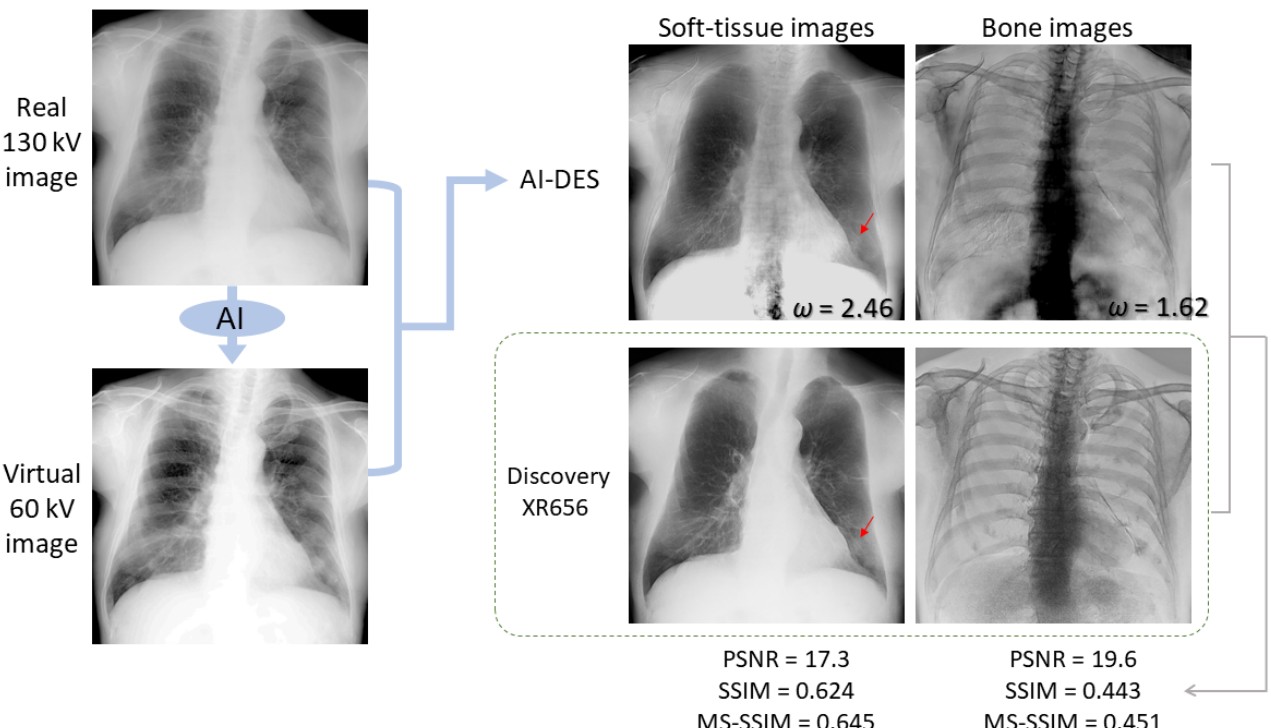

**Figure 6.** An example of soft-tissue- and bone-enhanced images generated by AI-DES with real 130 kV and virtual 60 kV images in comparison to the enhanced images generated by the Discovery XR656 system. The weight factors and similarity indices between the enhanced images generated by AI-DES and Discovery XR656 are presented. The red arrow indicates a pulmonary lesion.

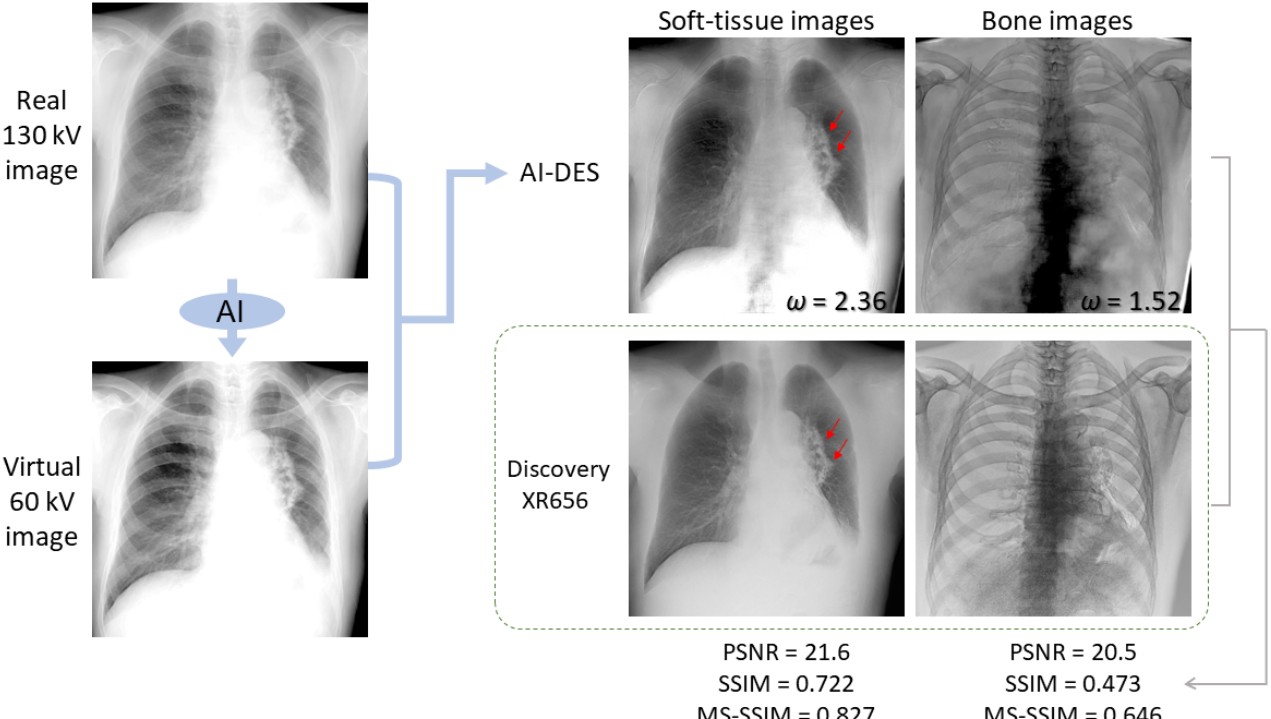

**Figure 7.** Another example of soft-tissue- and bone-enhanced images generated by AI-DES with real 130 kV and virtual 60 kV images in comparison to the enhanced images generated by the Discovery XR656 system. The weight factors and similarity indices between the enhanced images generated by AI-DES and Discovery XR656 are presented. The red arrows indicate a pulmonary lesion.

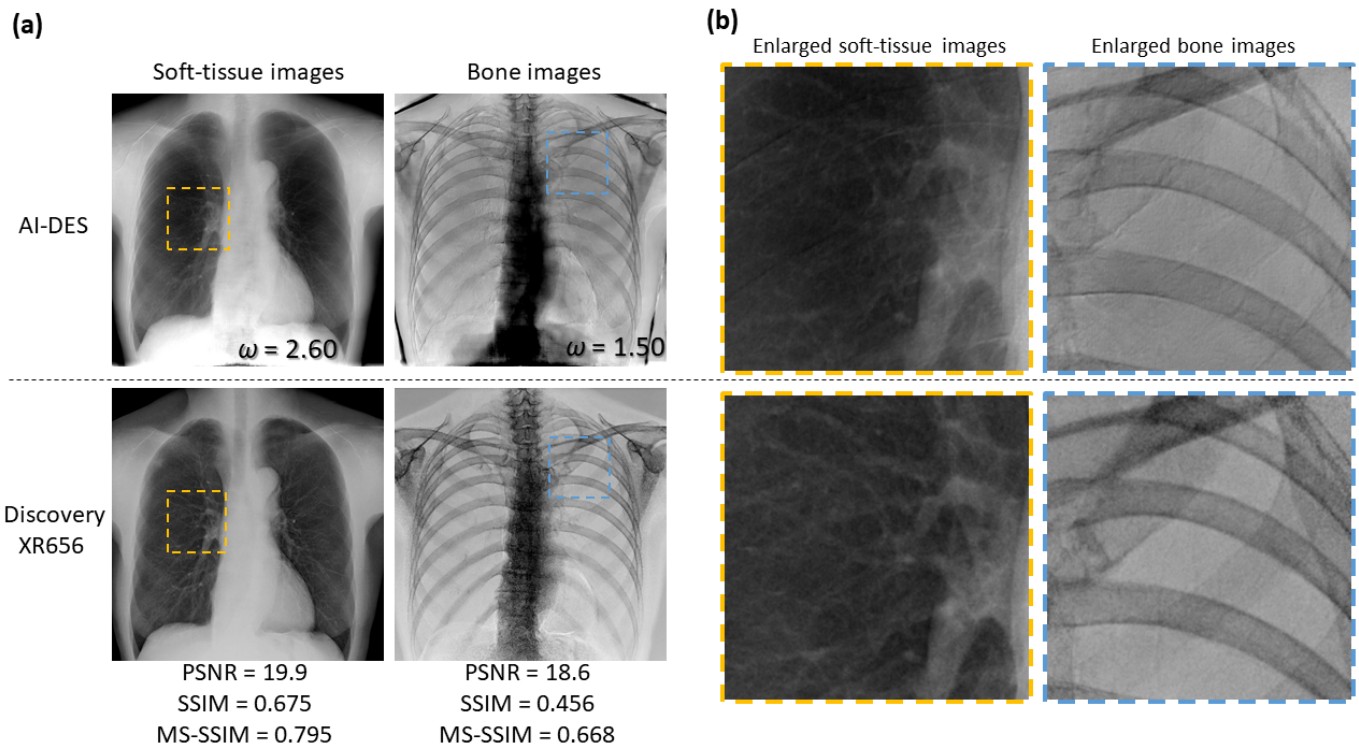

**Figure 8.** Comparison of soft-tissue- and bone-enhanced images obtained by AI-DES and Discovery XR656 for a test case. (**a**) Overall views of the enhanced images. The weight factors and similarity indices between the enhanced images generated the two systems are also presented. (**b**) Enlarged views of the specific areas are enclosed by the orange and blue dotted boxes in the overall views.

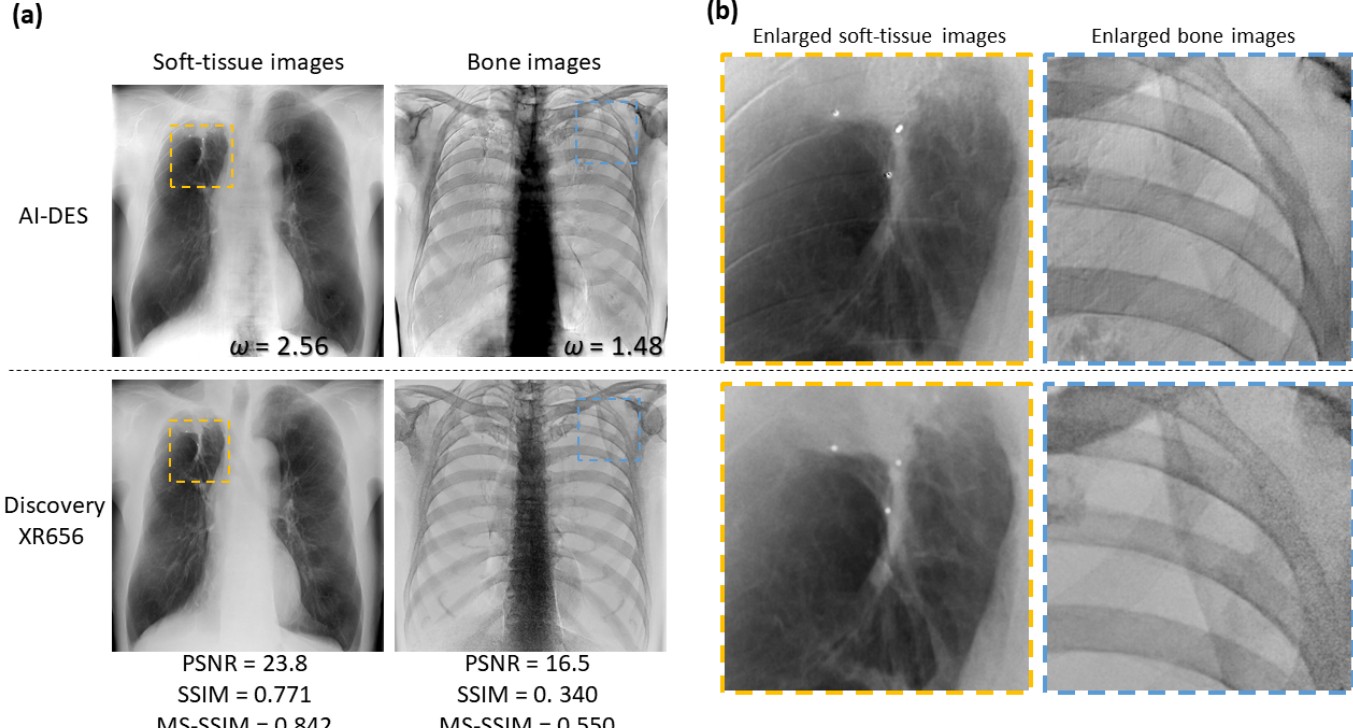

**Figure 9.** Comparison of soft-tissue- and bone-enhanced images obtained by AI-DES and Discovery XR656 for another test case. (**a**) Overall views of the enhanced images. (**b**) Enlarged views of specific areas are enclosed by orange and blue dotted boxes in the overall views.

## 4. Discussion

The main purpose of this study was to produce soft-tissue- and bone-enhanced images from only high-energy images by developing an AI-DES system. The performance of the AI model was assessed by image similarity between the generated and real low-energy images. The overall performance of the AI-DES system was evaluated by comparing the image quality of the soft-tissue- and bone-enhanced images with those produced by a clinically applied DES system.

As shown in Figure 4, our AI–DES successfully generated virtual low-energy images that closely resembled the real low-energy images. The image similarity indicated a high level of the indices (PSNR = 33.8 dB, SSIM = 0.984 and MS-SSIM = 0.957). Our system achieved this result without the need for a specific imaging detector with a metal plate or multiple X-ray exposures. Consequently, AI-DES has the potential to reduce the image noise and X-ray dose compared to existing DES systems. Additionally, its motion-artifact-free nature resulting from avoidance of multiple exposures makes AI-DES particularly valuable for elderly or critically ill patients who may have difficulty in holding their breath.

Furthermore, in the soft-tissue-enhanced images obtained via the weighted subtraction process, bone tissues within the lung fields were effectively suppressed, although faint residual bone edges or shadows were observed. We also confirmed that the contrast of pulmonary lesions was clearly enhanced in some cases, as indicated by red arrows in Figures 6 and 7. Nevertheless, the soft tissue contrast was slightly inferior to that of the clinical system, as shown in Figures 8b and 9b. Alternatively, AI-DES demonstrated advantages in terms of sharpness and noise characteristics. Overall, we subjectively verified that the image quality in the lung fields is almost comparable to that in the clinically applied system. However, these limited cases provide only weak evidence regarding the usefulness of AI-DES for improving lesion detectability. Therefore, further investigation is needed to determine whether AI-DES can truly improve the detectability for a larger number of cases.

The similarity indices (PSNR = 21.1 dB, SSIM = 0.711, and MS-SSIM = 0.794) for our soft-tissue-enhanced images were considerably lower than the results of existing AI models for bone suppression in chest radiography [23,25,28–31]. Zhou et al., generated bone-suppressed chest radiographs with a resolution of $256 \times 256$ using a cGAN-based model citeref-DcGANand reported an average PSNR of 35.5 dB and an SSIM of 0.975 for the similarity between the generated images and the ground truth. Rajaraman et al., also developed a CNN model called DeBoNet, which suppresses bones in chest radiographs with a resolution of $256 \times 256$ [30]. They achieved image similarity to the ground truth produced with commercial software, with an average PSNR of 36.8 dB, an SSIM of 0.947, and an MS-SSIM of 0.985. The low similarity indices of our soft-tissue-enhanced images can be attributed to bone edge artifacts and excessive contrast outside the lung fields, as well as the presence of black or white areas beyond the body contours where pixel values are nearly 0 or extremely high. Undoubtedly, our generated images need improvements by artifact reduction. However, a more substantially meaningful comparative analysis would have been provided by excluding the outer areas of the body contours for similarity index calculation. Additionally, it should be noted that our image generation was accomplished with a resolution of $1024 \times 1024$, which is certainly a greater challenge compared to $256 \times 256$ image synthesis.

On the other hand, the generated bone-enhanced images contained noticeable blacked-out artifacts on and around the spines, despite the successful suppression of soft tissues and the selective enhancement of ribs. The ribs and clavicles demonstrated significantly superior sharpness and noise characteristics compared to the images obtained by the Discovery system, as shown in Figures 8b and 9b. However, the visibility of the spines was much lower than that of the clinically applied system, primarily due to the presence of blacked-out artifacts. Considering the importance of selectively enhanced bone images for the detection of bone fractures or tumors, addressing this issue is crucial. In addition, some of the soft-tissue-enhanced images contained faint bone shadows, as previously mentioned. We attribute these issues to the following three factors.

First, the training dataset was limited to only 240 cases. Training the AI network on a larger image dataset may address these issues, although our AI network has already produced virtual low-energy images that closely resemble real images. In future studies, we intend to investigate whether the training dataset contains misalignments between high- and low-energy images owing to patient motion. Removing misaligned data may further improve the performance of the AI network.

Secondly, we wonder whether the weight factors used in the subtraction process were not optimized. We determined the individual values of $\omega$ for each patient to generate more selectively enhanced soft tissue and bone images. However, these values may not have been optimal. Moreover, bone thickness and density vary across different body locations, even within the same patient. Accordingly, it is possibly challenging to keep all bone shadows but completely eliminate soft tissues in the entire image with a single weight factor. Even so, we believe that selectively enhanced tissue images aid in lesion detection if target tissues were effectively enhanced. Therefore, we will attempt to selectively emphasize target tissues by optimizing the weight factor in future studies.

Thirdly, the effect of quantization error was perhaps more prominently visualized through log amplification in the subtraction process. Alternatively, the cause of the artifacts may be attributed to the discrepancy between the virtual and real low-energy images, despite their high similarity indices. We noticed that the location of black artifacts around lower thoracic and lumbar spines in bone-enhanced images corresponded to areas with excessively high X-ray absorption, where the pixel values ranged from 0 to 2 in real high-energy and virtual low-energy images. We speculate that numerical errors of such small values are further enhanced by logarithmic conversion, resulting in noticeable artifacts. We performed the subtraction process using 12 bit image data, but in future studies, we plan to use images with higher contrast resolution to eliminate artifacts.

Here, we discuss the computational complexity of our AI-DES. We adopted the widely used pix2pix with a few changes, so its implementation was not a challenging task. Despite using only 240 cases for training, the similarity between the produced and target images was high enough to support stable performance. Although the network requires paired data of high- and low-energy images, the preprocessing is far from a complex computation. However, due to the high resolution of the generated images ($1024 \times 1024$), the batch size had to be limited to a maximum of two to prevent an excessive GPU load. Next, the AI-DES requires the application of a weighted image subtraction process to the images generated by the AI network. In our current method, the weight factors are determined manually through visual inspection in order to emphasize the desired tissues to the greatest extent possible. Taken together, we consider that the construction of our AI network is no more complicated than those proposed in existing studies to directly generate bone-suppressed images [23–31]. However, the image subtraction process, which is the latter part of AI-DES, requires human intervention and time rather than computational complexity. While we also aim to automate the subtraction process in future work, we anticipate that it will involve complex computations, as attempted by Do et al. [19].

This study is also subject to some limitations, as described below. First, we used anonymized image data and did not include any patient information, such as gender, age, presence or absence of lesions, or medical history. Evaluating the performance separately according to various categories may provide useful insights for further improvements in this development. Particularly, a comparison of the performance between normal and diseased patients would be valuable. Next, a bias may have been introduced introduced in this study due to the use of image data collected at a single site using a specific imaging system. It is necessary to verify the performance with other datasets in future studies. Moreover, the image quality of the tissue-enhanced images generated by AI-DES was assessed subjectively by only three radiological technologists. Future studies should involve radiologists or thoracic physicians to evaluate the image quality more comprehensively.

To conclude this paper, we present one more points of superiority of the AI-DES system. It is possible to create selectively enhanced images of tissues with any linear

attenuation coefficient by adjusting the weight factor in the subtraction process. Although this initial development report was focused on generating soft-tissue- and bone-enhanced images, in future work, we aim to generate enhanced images targeting specific lesions for individual patients. This approach will be feasible due to the utilization of virtual low-energy images, since it differs from existing image processing approaches that directly create bone-suppressed images [23–31].

## 5. Conclusions

Our developed AI–DES successfully generated virtual low-energy images from high-energy images obtained in routine radiography. We demonstrated that the virtual low-energy images have a high similarity to real images. Additionally, the AI-DES achieved the production of soft-tissue- and bone-enhanced images through weighted subtraction processing. The soft-tissue-enhanced images showed comparable quality, especially within lung fields, to those produced by the existing DES system while avoiding difficulties such as increased noise and exposure dose increments. The bone-enhanced images showcased advantages in terms of sharpness and noise characteristics, although noticeable artifacts on and around lower thoracic and lumbar spines need to be addressed. In future work, we aim to improve the image quality, particularly in bone-enhanced image generation, by making modifications to the AI-DES. It is also essential to evaluate the performance by involving radiologists or thoracic physicians for a wide range of image cases.

**Author Contributions:** Conceptualization, T.I. and T.T.; methodology, T.T. and T.I.; software, T.T. and A.Y.; validation, A.K. and A.Y.; formal analysis, A.K.; investigation, A.K., A.Y. and T.T.; resources, M.S.; data curation, M.S.; writing—original draft preparation, A.K. and A.Y.; writing—review and editing, T.I.; visualization, A.K and A.Y.; supervision, T.I.; project administration, T.I. All authors have read and agreed to the published version of the manuscript.

**Funding:** This research received no external funding.

**Institutional Review Board Statement:** All images used in this study were approved by the Research Ethics Review Committee of Kitasato University Hospital and Osaka University Graduate School of Medicine.

**Informed Consent Statement:** A waiver of informed consent was approved by the Research Ethics Review Committee of Kitasato University Hospital and Osaka University Graduate School of Medicine because of the retrospective nature of this study. According to Japan's Ethical Guidelines for Medical and Health Research Involving Human Subjects, patients were given the opportunity to "opt out".

**Data Availability Statement:** Not applicable.

**Conflicts of Interest:** The authors declare no conflict of interest.

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
