# Peer review of "Development of Artificial Intelligence-Based Dual-Energy Subtraction for Chest Radiography"

_applsci, doi:10.3390/app13127220_

Round 1

Reviewer 1 Report

The problems with the paper are as follows:

1.Abstract did not clearly describe the main improvements and contributions of the method in this article, and it is recommended to make modifications.

2.Line 176,“The average PSNR and SSIM values across all test cases were 33.8 and 0.984”. The unit of PSNR is dB, please add it.

3.It is recommended to increase the comparison with the methods of the past three years to demonstrate the advantages and effectiveness of this method.

There is basically no problem with English.

Reviewer 2 Report

 Review Comments

The presented work explained artificial intelligence-based DES (AI–DES) technology for chest radiography to overcome these limitations. Using a trained pix2pix model on clinically acquired chest radiograph pairs, we successfully converted 130 kV images into virtual 60 kV images that closely resemble the real images. The averaged peak signal-to-noise ratio (PSNR) and structural similarity (SSIM) between virtual and real 60 kV images were 33.8 and 0.984, respectively.  However, the following major corrections can be considered by the authors to further improve the quality of the manuscript.

 I have some major corrections and suggestions below:-

1. Authors must show explain the novel contribution of the work with proper justification of the outcomes. What novelty is established in this work compared to existing works? Novel contribution of the work can be added at end of introductions with proper justification of the outcomes.

2. Literature survey is missing and need to be modified based on current state of art methods. Some more paper based on current study in chest radiography.

3. The computational complexity of the algorithm must be discussed. Also, compare the proposed method in terms of computational complexity?

4. Future work and limitations of the proposed work can be added and discussed.

5. Layers details of proposed architecture must be included.

6. Comparative analysis with respect to various performance metrics is missing? The comparison can be a bit unfair if different data is not used for comparative analysis.

7. Has the Author implemented the architecture from scratch and identified the novel condition in deep networks.

8. Various other performance metric need to be discussed and definitions of performance parameters must be included.

9. Visualized Results with respect to various categories of data sets must be discussed and presented.

10. Comparative analysis of various performance parameters with respect to sate of art methods must be discussed.

13. How much data should be considered for training and testing for architecture implementation? Details of training and testing data sets must be tabulated.

14. Specification of the implementation platform is missing.

Reviewer 3 Report

The work is concerned with the development of AI-driven, dual-energy subtraction (DES) system, based on the well-known Pix2Pix model, for chest radiography to address concerns related to increased noise and motion artifacts, associated with the use of one-shot and two-shot methods. It achieves successful conversion of 130 kV images into virtual 60 kV images, closely resembling real images.

Minor comments include:

Please highlight in the paper any modifications applied to the standard Pix2Pix model for the purpose of this application;

It could be interesting to segment the patients into healthy and diseased groups and present PSNR and SSIM statistics across the two groups;

Some additional analysis would helpful in further characterizing the origin of the artifacts present around the lower thoracic and lumbar spines;

While the performance supports the goal of improving the detectability of pulmonary lesions, further details would be useful in explaining future improvements to image quality.

Overall, the quality of the English language is very good. There are only minor typos, which could be easily resolved via spell checking.

Round 2

Reviewer 1 Report

Accept in present form

Reviewer 2 Report

All my concerns and comments has been added successfully. Accept it in current form.